# Acceptability of a dapivirine levonorgestrel vaginal ring in two Phase 1 trials (MTN-030/ IPM 041 and MTN-044/IPM 053/CCN019): Implications for multipurpose prevention technology development

Barbara A. Friedland[1]\*, Holly Gundacker[2], Sharon L. Achilles[3,4¤a], Beatrice A. Chen[3,4], Craig Hoesley[5], Barbra A. Richardson[6,7], Clifton W. Kelly[2], Jeanna Piper[8], Sherri Johnson[9], Brid Devlin[10], John Steytler[10], Kyle Kleinbeck[10], Bindi Dangi[10], Chantél Friend[10], Mei Song[3,4], Barbara Mensch[11], Ariane van der Straten[12], Cindy Jacobson[13¤b], Craig W. Hendrix[14], Jill Brown[15¤c], Diana Blithe[15], Sharon L. Hiller[3,4], on behalf of the MTN-030/IPM 041 and MTN-044/IPM 053/CCN019 Protocol Teams for the Microbicide Trials Network and the Contraceptive Clinical Trials Network[¶]

1 Center for Biomedical Research, Population Council, New York, NY, United States of America, 2 Statistical Center for HIV/AIDS Research and Prevention (SCHARP), Fred Hutchinson Cancer Center, Seattle, Washington, United States of America, 3 Department of Obstetrics, Gynecology, and Reproductive Sciences, University of Pittsburgh, Pittsburgh, Pennsylvania, United States of America, 4 Magee-Womens Research Institute, Pittsburgh, Pennsylvania, United States of America, 5 University of Alabama at Birmingham Heersink School of Medicine, Birmingham, Alabama, United States of America, 6 Vaccine and Infectious Disease Division, Fred Hutchinson Cancer Center, Seattle, Washington, United States of America, 7 Department of Biostatistics and Global Health, University of Washington, Seattle, Washington, United States of America, 8 National Institutes of Allergy and Infectious Disease, NIH, Bethesda, Maryland, United States of America, 9 FHI 360, Durham, North Carolina, United States of America, 10 International Partnership for Microbicides, Silver Spring, Maryland, United States of America, 11 Population Council, New York, NY, United States of America, 12 RTI International, San Francisco, CA, United States of America, 13 Department of Pharmaceutical Sciences, School of Pharmacy, University of Pittsburgh, Pittsburgh, Pennsylvania, United States of America, 14 Division of Clinical Pharmacology, Department of Medicine, Johns Hopkins University School of Medicine, Baltimore, Maryland, United States of America, 15 Contraceptive Development Program, DIPHR, National Institute of Child Health and Human Development, NIH, Bethesda, Maryland, United States of America

¤a Current address: Bill & Melinda Gates Foundation, Seattle, Washington, United States of America
¤b Current address: FHI360, Durham, NC, United States of America
¤c Current address: Uniformed Services University of the Health Sciences, Bethesda, Maryland, United States of America
¶ Membership of the MTN-030/IPM 041 and MTN-044/IPM 053/CCN019 Protocol Teams for the Microbicide Trials Network and the Contraceptive Clinical Trials Network is listed in the Acknowledgments.
\* bfriedland@popcouncil.org

**Data Availability Statement:** The datasets are available on the Population Council's Dataverse site

## Abstract

End-user feedback early in product development is important for optimizing multipurpose prevention technologies for HIV and pregnancy prevention. We evaluated the acceptability of the 90-day dapivirine levonorgestrel ring (DPV-LNG ring) used for 14 days compared to a dapivirine-only ring (DVR-200mg) in MTN-030/IPM 041 (n = 23), and when used for 90 days cyclically or continuously in MTN-044/IPM 053/CCN019 (n = 25). We enrolled healthy, non-pregnant, HIV-negative women aged 18–45 in Pittsburgh, PA and Birmingham, AL (MTN-

housed on the Harvard Dataverse at the following links: MTN-044: https://doi.org/10.7910/DVN/VZLUFP MTN-030: https://doi.org/10.7910/DVN/BYPUJC.

**Funding:** The author(s) received no specific funding for this work.

**Competing interests:** The following authors have read the journal's policy and have competing interests: Sharon L. Achilles has received consulting fees from Mayne Pharma and Merck and has received research funding from The National Institutes of Health, the US Food and Drug Administration, the Pennsylvania Department of Health, Society of Family Planning Research Fund, Estetra SRL (an affiliate company of Mithra Pharmaceuticals), EvoFem, and Merck, all of which were managed by Magee-Womens Research Institute. Barbra A. Richardson has received payment from Gilead Sciences for DSMB membership. Brid Devlin and John Steytler were full-time salaried employees of the International Partnership for Microbicides (IPM), a non-profit company registered in the United States of America, at the time the work was performed. Barbara A. Friedland was a full-time salaried employee of the Population Council, a non-profit company registered in the United States of America, at the time the work was performed. Craig W. Hendrix is an Inventor on a patent relating to vaginal microbicides and the founder of a microbicide development company, both unrelated to this study product and both managed by Johns Hopkins University. Beatrice A. Chen has served on a Merck & Co. advisory board and has received research grants from Sebela, Mylan, and Medicines360, all of which were managed by Magee-Women's Research Institute. NIH employees (Diana L. Blithe, Jill Brown, and Jeanna M. Piper) contributed to the study design, manuscript development and the decision to publish as well as providing safety oversight during study conduct but had no role in data collection and analysis. All other authors have declared that no competing interests exist. This does not alter our adherence to PLOS One policies on sharing data and materials.

030 only). Self-reports of vaginal bleeding and adherence (ring removals, expulsions) were collected via daily short message service. Acceptability data were recorded in face-to-face interviews at study exit. We assessed differences in acceptability by product characteristics and adherence; and associations between baseline characteristics/demographics, number of bleeding days, adherence, and overall acceptability. Most (21/23) women in the 14-day MTN-030 study and about half (13/25) in the 90-day MTN-044 study liked their assigned rings. In MTN-030 there were no significant associations between any variables and overall acceptability of either ring. In MTN-044, women who disliked the DPV-LNG ring had a significantly higher incidence of unanticipated vaginal bleeding, and reporting that vaginal bleeding changes were unacceptable than those who liked it. Although we found no overall association between adherence and acceptability, significantly more women who disliked (versus liked) the DPV-LNG ring reported expulsions during toileting. The DPV-LNG ring could meet the needs of women seeking simultaneous protection from HIV and unintended pregnancy. Addressing issues related to vaginal bleeding and expulsions early in product development will likely enhance acceptability of the DPV-LNG ring.

**Clinical Trial Registration:** MTN-030/IPM 041: ClinicalTrials.gov NCT02855346; MTN-044/IPM 053/CCN019: ClinicalTrials.gov NCT03467347.

## Introduction

Despite advances in HIV treatment and prevention over the last decade, AIDS-related diseases continue to be the leading cause of death globally among women aged 15–49 years old [1]. Oral pre-exposure prophylaxis (PrEP) is an HIV prevention option that is more than 90% effective in reducing HIV transmission when used consistently [2], yet most oral PrEP trials and demonstration projects in women have been plagued by poor uptake and adherence [3–8]. The dapivirine vaginal ring (DVR), which contains 25mg of the non-nucleoside reverse transcriptase inhibitor (NNRTI) dapivirine, is a novel HIV prevention product currently being rolled out in several African countries [9]. The DVR received a favorable opinion from the European Medicines Agency [10] based on two Phase 3 efficacy trials that demonstrated an approximate 30 percent HIV risk reduction [11, 12]. Secondary analyses and data from open label extension studies indicated that higher rates of adherence were correlated with increased protection [13–15]. Growing evidence suggests that many women may be more likely to use a multipurpose prevention technology (MPT) product that prevents pregnancy in addition to HIV infection [16–21]. Some participants in one of the Phase 3 trials of the DVR reported telling their partners they were using a new contraceptive rather than an HIV prevention product, a strong indication that pairing HIV prevention with contraception may facilitate PrEP use [22]. Several novel contraceptive MPTs are in development, including a 90-day ring containing dapivirine for HIV prevention and the progestin levonorgestrel for pregnancy prevention [23–26].

Dapivirine (DPV) formulated as vaginal rings, gels, vaginal films, and in an oral formulation has been evaluated in multiple clinical trials and found to be safe [27]. Levonorgestrel (LNG) is a synthetic progestin that has been approved for use in contraceptive products for more than 30 years [28] and is now being developed in several MPT vaginal rings [29, 30]. The higher dapivirine loading dose (200 mg vs 25 mg) in the DPV-LNG ring compared to the DVR was designed to enable less frequent ring replacements (quarterly versus monthly),

which may reduce user burden, facilitate streamlined service delivery, and thus, improve acceptability and adherence. Given the observed critical role of adherence in effectiveness for the DVR, it is important to identify modifiable factors that can impact correct and consistent use as early as possible in product development of the DPV-LNG ring [31–35]. In this paper, we present acceptability data from two Phase 1 trials of the DPV-LNG ring used continuously for 14 days (MTN-030/IPM 041) or either cyclically or continuously for 90 days (MTN-044/IPM 053/CCN019) [36].

## Methods

### Trial designs

MTN-030/IPM 041 (herein, referred to as MTN-030) was a Phase I, two-arm, double-blind, randomized trial conducted in 24 healthy, HIV-uninfected, non-pregnant women 18 to 45 years old at the University of Pittsburgh (Pittsburgh, PA) and the University of Alabama (Birmingham, AL). Methods have been described in detail previously [36]. Briefly, eligibility criteria included use of an effective, non-hormonal contraceptive (including sterilization and non-hormonal intrauterine devices [IUDs], but not condoms) or engaging exclusively in sex with women; regular menstrual cycles (21–35 days); and willingness to abstain from sexual activity and insertion of any vaginal products 24 hours before enrollment through the last day of product use. Participants were enrolled and randomized 1:1 to use one of two silicone elastomer rings for 14 days continuously: the DPV-LNG ring (200 mg of DPV, 320 mg of LNG) or a ring with dapivirine only–the DVR-200mg (200 mg of dapivirine). Both products are flexible, smooth, off-white rings with an outer diameter of 56mm and a cross-sectional diameter of 7.7mm, designed to provide sustained drug release over a minimum of three months [27, 36]. Participants' menstrual cycles were taken into consideration when scheduling the Enrollment Visit so that, ideally, no bleeding would occur during the 14 days of product use. The ring was inserted at Enrollment and removed on Day 14 (or an Early Termination Visit) by the participant herself (and checked for proper placement by a study staff member) or by a study staff member, if necessary (Fig 1). Participants were given detailed instructions on insertion and removal procedures, and what to do in the event of ring expulsion or loss. Primary and secondary objectives were safety, pharmacokinetics (PK), and reported vaginal bleeding during use of the DPV-LNG ring versus the DVR-200mg. Exploratory objectives included assessments of acceptability, based on self-reported attitudes about ring attributes and willingness to use it in the future, and adherence, based on the frequency and duration of voluntary removals and involuntary expulsions.

MTN-044/IPM 053/CCN019 (herein referred to as MTN-044) was a Phase I, two-arm, open-label, single-site, randomized trial of the DPV-LNG ring in 25 healthy, HIV-uninfected 18-45-year-old cis-gender women at the University of Pittsburgh. Eligibility criteria were similar to MTN-030, with the addition of consistent and correct male condom use as an acceptable contraceptive method and willingness to abstain from sex and vaginal product use for 24

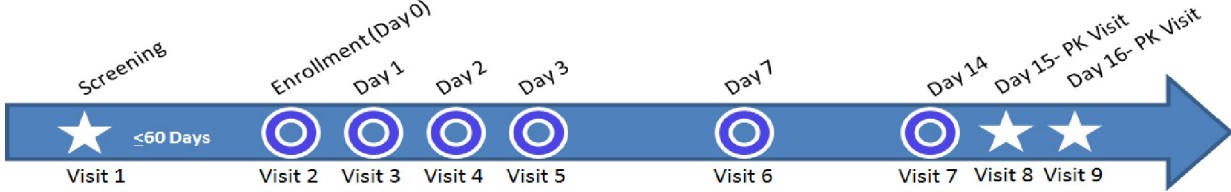

**Fig 1. Study schema for MTN-030/IPM 041.**

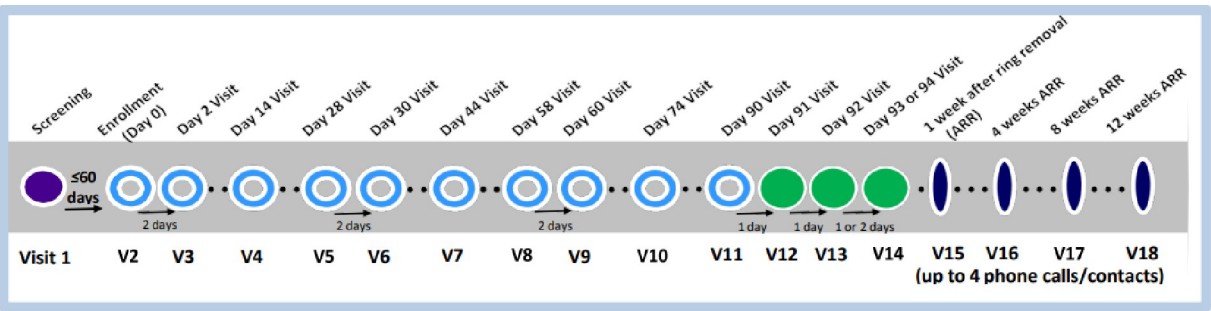

**Fig 2. Study schema for MTN-044/IPM 053/CCN019.**

hours before Enrollment, visits when samples were taken, and for one week after cervical biopsy collection (Fig 2). Participants were randomized 1:1 at Enrollment to use the DPV-LNG ring (200 mg DPV, 320 mg LNG) for 90 days continuously or cyclically (keep ring in place for 28 days; then wash and store for two days, prior to reinsertion to start the next cycle). Similar to MTN-030, participants were given detailed instructions on ring insertion and removal, and what to do in the event of ring expulsion or loss. The ring was inserted by participants (with a clinician confirming proper placement) or by a study staff member, if requested. For participants in the cyclic use group, rings were collected on Days 28 and 58, rinsed with tap water, stored at the study site for two days, and then reinserted on Days 30 and 60. Primary and secondary objectives were PK and safety of the DPV-LNG ring used continuously versus cyclically for 90 days, respectively. Exploratory objectives included assessments of vaginal bleeding patterns (number of episodes and total number of days of vaginal bleeding), acceptability (self-reported attitudes about ring attributes, use regimens and tolerability), and adherence (frequency and duration of voluntary ring removals and involuntary expulsions). Methods have been described in detail previously [36].

## Measures

**Acceptability.**   In both studies, we based our assessments on the Mensch et al. PrEP/Microbicide acceptability framework (Fig 3), which outlines specific domains and items to assess depending on clinical trial phase [37] and focuses on individual characteristics (influencing factors), product characteristics and use attributes (product acceptability), and execution (adherence). Quantitative acceptability data were collected via face-to-face interviews and recorded on case report forms (CRFs) at Enrollment (Visit 2 for both studies) and after completing ring use (Visit 7 in MTN-030 and Visit 11 in MTN-044). In MTN-030, there were two open-ended questions at the end of the interview in which participants were asked for any additional comments about the study or their experiences using the ring. In MTN-044, detailed acceptability data were also captured qualitatively via in-depth interviews and are presented elsewhere [38].

Baseline interviews included questions about influencing factors, such as previous use of contraception and vaginal products, sexual activity, and partnership status, as well as their thoughts and concerns prior to ring use. After completing product use (Day 14 or Day 90), questions about product acceptability emphasized product characteristics and use attributes, such as experiences inserting and removing the rings, comfort, and awareness of the ring during routine activity and during sex, partners' experiences, and side effects. The primary acceptability outcome in both trials was overall rating of the ring on Day 14 (MTN-030) or Day 90 (MTN-044) on a four-point Likert scale: dislike very much, dislike, like, and like very much.

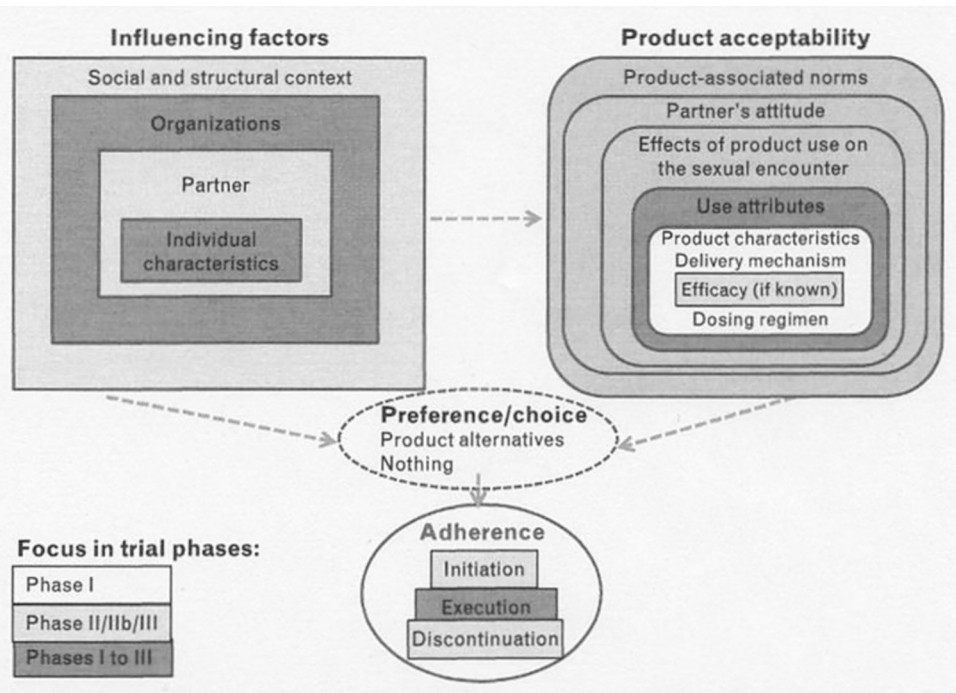

**Fig 3. Framework for evaluating acceptability in microbicide and PrEP trials [37].**

*Vaginal bleeding.* Participants in both trials received a daily short message service (SMS) in which they were asked if they had experienced any spotting or bleeding in the last 24 hours and, if so, if it was light, moderate, or heavy. SMS has been used previously to track bleeding patterns and adherence in research and in service delivery [39–42]. In MTN-044, women who reported any bleeding were also asked how bothersome the spotting or bleeding had been (not at all, a little, somewhat, very much). Participants in MTN-030 were also interviewed about bleeding at every study visit (Visits 3 through 9). In MTN-044, participants' daily SMS entries were transcribed onto a Bleeding SMS CRF at each follow-up visit (Visits 3 through 14).

**Adherence.** In both trials, adherence was measured based on ring outages, including the frequency and duration of expulsions (ring falling out) or unscheduled removals. In MTN-030, adherence was assessed using multiple methods [43, 44], including daily SMS, interviews recorded on CRFs at every study visit during the product use period (Visits 3, 4, 5, 6 and 7), and data convergence interviews at Visits 6 (Day 7) and 7 (Day 14). SMS captured the frequency of partial or full outages, how the ring came out (deliberate removal or expulsion), if applicable, and if an outage lasted more than three hours. The Ring Adherence CRF captured dates and times of any outages and reasons for removals or expulsions. The Ring Outage Convergence Interview was only used to clarify any discrepancies between data collected via SMS and on the CRFs. The SMS and CRF data, along with the counselor's/clinician's converged assessment of the most likely ring adherence, constituted the summary adherence database. In MTN-044, adherence was assessed via weekly SMS messages, as described for MTN-030 with additional questions about the circumstances in which the ring came out (such as during menses or before, during or after sex).

## Statistical analysis

Our original plan was to examine differences in acceptability, adherence, and bleeding patterns by study arm (DPV-LNG ring vs DVR-200mg; or DPV-LNG ring used cyclically or continuously). We assessed differences between study product arms by end-of-study product preferences and acceptability using Chi-squared or Fisher's exact tests for categorical variables and the Mann-Whitney U test for continuous variables. However, we found no significant differences in any outcomes by arm in either study. Therefore, we focused our analysis on factors that affected overall acceptability, such as baseline characteristics, vaginal bleeding, and adherence, and used aggregate data combining study arms for all analyses. We dichotomized overall acceptability into "like," which included "like" and "like very much" versus "dislike," which included "dislike" and "dislike very much." We calculated vaginal bleeding incidence rates as the total number of bleeding days per person-month and present the 95% confidence interval of the bleeding incidence rates. We classified participants as fully adherent if they reported having kept the ring inserted all the time during the study, without any product discontinuations or holds, and without any unscheduled ring outages (removals or expulsions). We used descriptive statistics to summarize overall acceptability by baseline sociodemographic characteristics and sexual behavior, and by specific aspects of acceptability (such as product characteristics or use attributes) reported at the last product use visit (Day 14 or Day 90). We assessed differences in overall acceptability by product characteristics and adherence using Chi-squared or Fisher's exact tests for categorical variables. We used logistic regression models to test associations between participant baseline characteristics/demographics, bleeding days, study product adherence, and overall acceptability. In the multivariable logistic regression model, we used stepwise selection to retain only the baseline factors significant at the 0.10 level in the final adjusted model. Due to the exploratory nature of these analyses, we made no adjustments for multiple comparisons. We used SAS® 9.4 (SAS Institute, Inc., Cary, NC) and R software (version 4.1.3) to generate all analyses.

## Ethics statement

The MTN-030 study protocol was approved by the local Institutional Review Boards at the University of Alabama and the University of Pittsburgh and was overseen by the regulatory infrastructure of the Division of AIDS (DAIDS) and Microbicide Trials Network (MTN). The MTN-044 study protocol was approved by the local Institutional Review Board at the University of Pittsburgh and was overseen by the regulatory infrastructure of the Contraceptive Clinical Trials Network and the MTN. All participants provided written informed consent before undergoing any procedures.

## Results

As shown in Table 1, 23 participants were eligible, enrolled, and completed MTN-030 (May 3-August 18, 2017). As previously described [36], 24 participants were enrolled and completed the study, however, one participant in the DVR-200mg arm was found to be ineligible after completing the study and was excluded from analyses. In MTN-044, 21/25 participants who enrolled (July 17, 2018-October 7, 2019), completed the study. Four participants withdrew early: one due to recurrent ring expulsions, one who used HIV post-exposure prophylaxis (a protocol-prohibited medication) following sexual assault, one who experienced an unrelated adverse event (AE), and one who did not agree to the cervical biopsy procedure. We included data in our analyses for all 25 participants enrolled through the point of discontinuation. Most participants in both studies were White, non-Hispanic, college-educated, in their 30's, with a

**Table 1. Baseline characteristics by study and study arm (MTN-030/IPM 041 [n = 23] and MTN-044/IPM 053/CCN019 [n = 25]).**

| Characteristic | MTN-030/IPM 041 | | MTN-044/IPM 053/CCN019 | |
|---|---|---|---|---|
| | **14-day** | **14-day** | **90-day** | **90-day** |
| | **DVR-200mg (n = 11)** | **DPV-LNG ring (n = 12)** | **DPV-LNG ring continuous (n = 12)** | **DPV-LNG ring cyclic (n = 13)** |
| Mean age, years (min-max) | 31 (22–41) | 33 (20–43) | 32 (22–43) | 37 (21–43) |
| Race | | | | |
| Black | 2 (18%) | 5 (42%) | 2 (17%) | 1 (8%) |
| White | 8 (73%) | 6 (50%) | 8 (67%) | 12 (92%) |
| Asian | 0 (0%) | 0 (0%) | 2 (17%) | 0 (0%) |
| Mixed race[1] | 1 (9%) | 1 (9%) | 0 (0%) | 0 (0%) |
| Ethnicity | | | | |
| Hispanic or Latino | 0 | 0 | 1 (8%) | 1 (8%) |
| Not Hispanic or Latino | 11 (100%) | 12 (100%) | 11 (92%) | 12 (92%) |
| Marital status | | | | |
| Single, never married | 8 (74%) | 9 (75%) | 10 (83%) | 8 (62%) |
| Married/in relationship | 3 (27%) | 3 (25%) | 1 (8%) | 5 (38%) |
| Divorced | 0 (0%) | 0 (0%) | 1 (8%) | 0 (0%) |
| Primary sex partner | | | | |
| Man | 7 (64%) | 4 (33%) | 5 (42%) | 8 (62%) |
| Woman | 2 (18%) | 1 (8%) | 1 (8%) | 0 |
| Other[2] | 0 | 1 (8%) | 0 | 1 (8%) |
| None | 2 (18%) | 6 (50%) | 6 (50%) | 4 (31%) |
| Highest level of education completed | | | | |
| High school | 2 (18%) | 6 (50%) | 4 (33%) | 4 (31%) |
| College | 9 (82%) | 6 (50%) | 8 (67%) | 9 (69%) |
| $\geq$ 1 full-term live birth | 5 (45%) | 5 (42%) | 4 (33%) | 5 (38%) |
| Previous contraceptive use | | | | |
| Male condom | 11 (100%) | 10 (83%) | 12 (100%) | 13 (100%) |
| Oral contraceptive | 9 (82%) | 11 (92%) | 7 (58%) | 8 (62%) |
| Emergency contraception | 5 (45%) | 2 (17%) | 9 (75%) | 2 (15%) |
| Vaginal ring | 3 (27%) | 4 (33%) | 0 (0%) | 4 (31%) |
| Intrauterine device (IUD) | 4 (36%) | 6 (50%) | 4 (33%) | 4 (31%) |
| Other hormonal[3] | 3 (27%) | 3 (25%) | 3 (25%) | 6 (46%) |
| Other[4] | 10 (91%) | 10 (83%) | 9 (75%) | 9 (69%) |
| Douched (ever) | 3 (27%) | 3 (25%) | 4 (33%) | 2 (15%) |
| Ever had vaginal sex | 11 (100%) | 11 (92%) | 12 (100%) | 13 (100%) |
| Used condom at last vaginal sex | | | | |
| Yes, male condom | 5 (45%) | 3 (25%) | 7 (58%) | 6 (46%) |
| No | 6 (55%) | 8 (67%) | 5 (42%) | 7 (54%) |
| Not applicable | 0 (0%) | 1 (8%) | 0 (0%) | 0 (0%) |
| Ever had anal sex | 8 (73%) | 2 (17%) | 8 (67%) | 4 (31%) |
| Total number lifetime sexual partners | | | | |
| Median (IQR) | 10.0 (5.0,12.0) | 5.0 (3.0, 11.5) | 8.5 (8.0, 11.5) | 5.0 (3.0, 7.0) |
| Min-max | 1, 25 | 2, 40 | 1, 35 | 1, 32 |

1. "Mixed race" includes one American Indian/Alaska Native and white and one Asian and white

2. "Other" includes Transgender Female (MTN-030, "male to female transgender") and Transgender Male (MTN-044, "Participant's partner is a transgender man").

3. Includes patch, injectable, implant.

4. Includes spermicide, withdrawal, female condom, fertility awareness, sterilization, and "Other." In MTN-030, abstinence and sterilization were included as text in "Other, specify" and were not consistently collected for all participants.

primary partner, and without children. Participants had a wide range of prior experiences with contraceptive use and numbers of lifetime sexual partners.

## Acceptability

Table 2 displays opinions about overall acceptability and specific product characteristics by study arm for both trials. As noted above, we found no significant differences by arm in either study.

## MTN-030

In MTN-030, 21/23 participants (91.3%; 95% CI 72.0%-98.9%) participants said they liked their assigned ring at the end of the study; the two women who disliked the ring were both in the DPV-LNG ring arm. Most participants said they liked the ring the same (74%) or more (22%) than they thought they would before they had used it, with no differences by study arm. All 23 women said their assigned ring was easy to use and that it was comfortable to wear continuously for 14 days, the majority of whom (65%) said it was easy to insert the first time they tried doing so. A majority (65%) reported having thought about the ring being in their bodies at some point during the study and 26% said they were aware of the ring during daily activities.

Nine women (DVR-200mg arm: 4/11, DPV-LNG ring arm: 5/12) reported that the ring increased the sensation of vaginal wetness, six of whom said the increased wetness bothered them a little and one who said it bothered her somewhat. Among women who did not report increased wetness, four (all in the DPV-LNG ring arm) reported increased dryness; two said the increased dryness did not bother them at all, one said it bothered her a little, and one said it bothered her somewhat. When asked if they would be willing to use a ring in the future like the one used in the trial, 63% of participants in the DVR-200mg arm and 50% of participants in the DPV-LNG ring arm said yes (Fisher's exact p = 0.68). Regardless of which ring they had used, the majority (74%) of participants said they preferred a ring combining HIV and pregnancy prevention versus separate methods for each.

When asked if they had any additional comments at the end of the study, 14 women had none and four re-emphasized that their assigned ring had been easy to use and comfortable or unnoticeable. Five participants had comments or concerns: two (1 DVR-200mg, 1 DPV-LNG ring) said that they could not imagine having sex with the ring in place; one participant (DPV-LNG ring) described the ring as "floppy" when trying to insert it, and said that although it was comfortable once it was in place, she believed it would be difficult to remove (the clinician removed it, per protocol); one participant (DPV-LNG ring) attributed early onset of menses to the ring and said she did not care for wearing it while menstruating; and one (DPV-LNG ring) reported becoming aroused during a movie and experiencing a burning sensation inside her vagina, which had never happened before.

## MTN-044

In MTN-044, 13/25 women (52.0%, 95% CI 31.3%-72.2%) reported liking the DPV-LNG ring, with no difference between those in the continuous (6/12) or cyclical use (7/13) arms (Table 2). Most participants found the ring easy to use (84%), easy to insert the first time they tried doing so (84%), and comfortable to wear for 90 days continuously or cyclically (84%). Most participants reported that they had thought about the ring being in their bodies during the study (88%) or that they were aware of the ring during daily activities (76%), with no significant differences by arm. Thirteen women (5/12, continuous arm; 8/13, cyclic arm) reported that the ring made their vaginas feel wetter. Six women said the increased vaginal wetness

**Table 2. Acceptability by study (MTN-030/IPM 041 [n = 23] and MTN-044/IPM 053/CCN019 [n = 25]) and by study arm.**

| Study: | MTN-030/IPM 041 | | | MTN-044/IPM 053/CCN019 | | |
|---|---|---|---|---|---|---|
| | DVR (n = 11) | DPV-LNG IVR (n = 12) | Total (n = 23) | Continuous (n = 12) | Cyclic (n = 13) | Total (n = 25) |
| Overall Acceptability: How much do you like the ring overall? | | | | | | |
| Dislike very much | 0 (0%) | 1 (8.3%) | 1 (4.4%) | 0 (0%) | 0 (0%) | 0 (0%) |
| Dislike | 0 (0%) | 1 (8.3%) | 1 (4.4%) | 6 (50.0%) | 6 (46.2%) | 12 (48.0%) |
| Like | 5 (45.5%) | 9 (75.0%) | 14 (60.9%) | 3 (25.0%) | 6 (46.2%) | 9 (36.0%) |
| Like very much | 6 (54.6%) | 1 (8.3%) | 7 (30.4%) | 3 (25.0%) | 1 (7.7%) | 4 (16.0%) |
| How worried are you about having the ring inside you? (baseline) | | | | | | |
| Somewhat worried | 1 (9.1%) | 1 (8.3%) | 2 (8.7%) | 1 (8.3%) | 4 (30.8%) | 5 (20.0%) |
| A little worried | 4 (36.4%) | 2 (16.7%) | 6(26.1%) | 3 (25%) | 5 (38.5%) | 8 (32.0%) |
| Not at all worried | 6 (54.6%) | 9 (75.0%) | 15 (65.2%) | 8 (66.7%) | 4 (30.8%) | 12 (48.0%) |
| How much do you like the ring overall? (baseline)[1] | | | | | | |
| Like/Like very much | N/A | N/A | N/A | 9 (75.0%) | 10 (76.9%) | 19 (76.0%) |
| Dislike | | | | 3 (25.0%) | 3 (23.1%) | 6 (24.0%) |
| Overall, how easy or difficult was it to use the ring? | | | | | | |
| Very difficult | 0 (0%) | 0 (0%) | 0 (0%) | 0 (0%) | 0 (0%) | 0 (0%) |
| Difficult | 0 (0%) | 0 (0%) | 0 (0%) | 1 (8.3%) | 3 (23.1%) | 4 (16.0%) |
| Easy | 1 (9.1%) | 5 (41.7%) | 6 (26.1%) | 2 (16.7%) | 6 (46.2%) | 8 (32.0%) |
| Very easy | 10 (90.9%) | 7 (58.3%) | 17 (73.9%) | 9 (75.0%) | 4 (30.8%) | 13 (52.0%) |
| The first time you inserted the ring in your vagina, was it difficult or easy to insert? | | | | | | |
| Very difficult | 0 (0%) | 0 (0%) | 0 (0%) | 0 (0%) | 0 (0%) | 0 (0%) |
| Difficult | 2 (18.2%) | 3 (25.0%) | 5 (21.7%) | 3 (25.0%) | 3 (23.1%) | 6 (24.0%) |
| Easy | 5 (45.5%) | 5 (41.7%) | 10 (43.5%) | 7 (58.3%) | 4 (30.8%) | 11 (44.0%) |
| Very easy | 2 (18.2%) | 3 (25.0%) | 5 (21.7%) | 2 (16.7%) | 6 (46.2%) | 8 (32.0%) |
| I never inserted the ring | 2 (18.2%) | 1 (8.3%) | 3 (13.0%) | 0 (0%) | 0 (0%) | 0 (0%) |
| Overall, how often did you think about the ring being inside your body? | | | | | | |
| Never | 4 (36.4%) | 4 (33.3%) | 8 (34.8%) | 2 (26.7%) | 1 (7.7%) | 3 (12.0%) |
| Some of the time | 7 (63.6%) | 7 (58.3%) | 14 (60.9%) | 9 (75.0%) | 8 (61.5%) | 17 (68.0%) |
| Most of the time | 0 (0%) | 0 (0%) | 0 (0%) | 1 (8.3%) | 4 (30.8%) | 5 (20.0%) |
| All of the time | 0 (0%) | 1 (8.3%) | 1 (4.4%) | 0 (0%) | 0 (0%) | 0 (0%) |
| Overall, were you aware of the ring during your normal daily activities? | | | | | | |
| Never | 8 (72.7%) | 9 (75.0%) | 17 (73.9%) | 4 (33.3%) | 2 (15.4%) | 6 (24.0%) |
| Some of the time | 3 (27.3%) | 2 (16.7%) | 5 (21.7%) | 8 (66.7%) | 9 (69.2%) | 17 (68.0%) |
| Most of the time | 0 (0%) | 1 (8.3%) | 1 (4.4%) | 0 (0%) | 2 (15.4%) | 2 (8.0%) |
| All of the time | 0 (0%) | 0 (0%) | 0(0%) | 0 (0%) | 0 (0%) | 0 (0%) |
| Overall, how did it feel to have the ring inside you every day? | | | | | | |
| Very comfortable | 7 (63.6%) | 5 (41.7%) | 12 (52.2%) | 5 (41.7%) | 2 (15.4%) | 7 (28.0%) |
| Comfortable | 4 (36.4%) | 7 (58.3%) | 11 (47.8%) | 6 (50.0%) | 8 (61.5%) | 14 (56.0%) |
| Uncomfortable | 0 (0%) | 0 (0%) | 0 (0%) | 1 (8.3%) | 3 (23.1%) | 4 (16.0%) |
| Very uncomfortable | 0 (0%) | 0 (0%) | 0 (0%) | 0 (0%) | 0 (0%) | 0 (0%) |
| Ever checked to see if the ring was still inside you? | 2 (18.2%) | 4 (33.3%) | 6 (26.1%) | 6 (50%) | 7 (53.8%) | 13 (52%) |
| Overall, ever noticed that vagina was wetter | 4 (36.4%) | 5 (41.7%) | 9 (39.1%) | 5 (41.7%) | 8 (61.5%) | 13 (52%) |
| If yes, how much has your vagina being wetter bothered you? | | | | | | |
| Not at all | 1 (25.0%) | 1 (20.0%) | 2 (22.2%) | 2 (40%) | 4 (50%) | 6 (46.2%) |
| A little | 2 (50.0%) | 4 (80.0%) | 6 (66.7%) | 2 (40%) | 4 (50%) | 6 (46.2%) |
| Somewhat | 1 (25.0%) | 0 (0%) | 1 (11.1%) | 0 (0%) | 0 (0%) | 0 (0%) |
| Very much | 0 (0%) | 0 (0%) | 0 (0%) | 1 (20%) | 0 (0%) | 1 (7.7%) |
| Overall, ever noticed that vagina was drier | 0 (0%) | 4 (57.1%) | 4 (28.6%) | 1 (8.3%) | 0 (0%) | 1 (4%) |

*(Continued)*

**Table 2.** (Continued)

| Study: | MTN-030/IPM 041 | | | MTN-044/IPM 053/CCN019 | | |
|---|---|---|---|---|---|---|
| | DVR (n = 11) | DPV-LNG IVR (n = 12) | Total (n = 23) | Continuous (n = 12) | Cyclic (n = 13) | Total (n = 25) |
| If yes, how much has your vagina being drier bothered you? | | | | | | |
| Not at all | 0 (0%) | 2 (50.0%) | 2 (50.0%) | 0 (0%) | 0 (0%) | 0 (0%) |
| A little | 0 (0%) | 1 (25.0%) | 1 (25.0%) | 0 (0%) | 0 (0%) | 0 (0%) |
| Somewhat | 0 (0%) | 1 (25.0%) | 1 (25.0%) | 1 (100%) | 0 (0%) | 1 (100%) |
| Very much | 0 (0%) | 0 (0%) | 0 (0%) | 0 (0%) | 0 (0%) | 0 (0%) |
| Overall, how acceptable were any changes in your bleeding pattern while using the ring?[1] | | | | | | |
| Acceptable[2] | N/A | N/A | N/A | 9 (75.0%) | 9 (69.2%) | 18 (72.0%) |
| Unacceptable | | | | 3 (25.0%) | 4 (30.8%) | 7 (28.0%) |

1. Question was not included in the questionnaire for MTN-030.

2. "Acceptable" includes those who reported no change in bleeding pattern.

bothered them a little, six said it bothered them somewhat and one said it bothered her a lot. One participant (continuous arm) reported increased dryness that she said bothered her somewhat.

## Vaginal bleeding

The heatmap (Fig 4) summarizes the number of bleeding days that occurred in MTN-030. As expected, most participants (15/23) had no vaginal bleeding during the trial. Four participants reported light or moderate menstrual bleeding (three in the DVR-200mg arm and one in the DPV-LNG ring arm) and four participants in the DPV-LNG ring arm reported light bleeding that was not associated with menses with no significant difference in vaginal bleeding incidence (bleeding days per person-month) between the DVR-200mg (1.3; 95% CI: 0.6, 2.5) and DPV-LNG (3.7; 95% CI: 2.4, 5.5) arms (Table 3).

In MTN-044, in which vaginal bleeding was expected, the bleeding patterns were similar for women in the continuous and cyclic use arms, as illustrated in the heatmaps (Fig 5). One participant (cyclic use arm) reported no bleeding during more than 91 days of follow-up, whereas heavy bleeding was reported by six participants, two in the cyclic arm and four in the

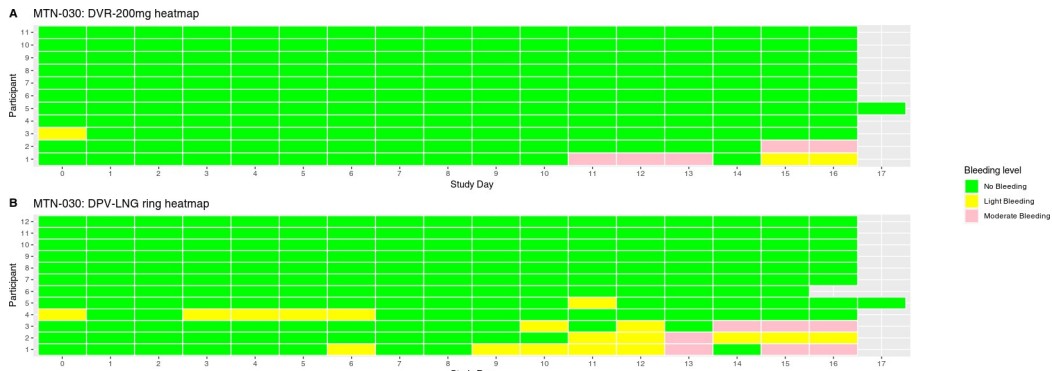

**Fig 4. Heatmap of bleeding patterns, MTN-030/IPM 041 (n = 23).** green = no bleeding, yellow = light bleeding, pink = moderate bleeding, red = heavy bleeding.

**Table 3. Adherence (expulsions and removals) and bleeding incidence by study (MTN-030/IPM 041 [n = 23] and MTN-044/IPM 053/CCN019 [n = 25]) and by study arm.**

| Study: | MTN-030/IPM 041 | | | MTN-044/IPM 053/CCN019 | | |
|---|---|---|---|---|---|---|
| | DVR (n = 11) | DPV-LNG IVR (n = 12) | Total (n = 23) | Continuous (n = 12) | Cyclic (n = 13) | Total (n = 25) |
| Adherent[1] | 11 (100%) | 9 (75.0%) | 20 (87.0%) | 7 (58.3%) | 6 (46.2%) | 13 (52.0%) |
| Number of **full** outages (expulsions or unscheduled removals) reported | | | | | | |
| 0 | 11 (100%) | 9 (75%) | 20 (87%) | 7 (58.3%) | 6 (46.2%) | 13 (52.0%) |
| 1 | 0 | 3 (25%) | 3 (13%) | 1 (8.3%) | 3 (23.1%) | 4 (16.0%) |
| 2 or more | 0 | 0 | 0 | 4 (33.3%) | 4 (30.8%) | 8 (32.0%) |
| Number of participants with ≥ 1 partial expulsion (partially came out/slipped out of place) | N/A | N/A | N/A | 7 (58.3%) | 12 (92.3%) | 19 (76.0%) |
| Number of participants with ≥ 1 complete expulsion (fully fell out) | 0 | 3 (25%) | 3 (13%) | 4 (33.3%) | 6 (46.2%) | 10 (40.0%) |
| Number of participants with ≥ 1 unscheduled ring removal | 0 | 0 | 0 | 3 (25.0%) | 4 (30.8%) | 7 (28.0%) |
| Number of ring outages, median (range) | 0 (0, 0) | 0 (0, 1) | 0 (0, 1) | 0 (0, 5) | 1.0 (0, 39) | 0 (0, 39) |
| Reasons for outages | | | | | | |
| Ring out of place/moved | 0 | 0 | 0 | 4 (33.3%) | 5 (38.5%) | 9 (36.0%) |
| Physical discomfort | 0 | 0 | 0 | 1 (8.3%) | 3 (23.1%) | 4 (16.0%) |
| Menses | 0 | 0 | 0 | 2 (16.7%) | 1 (7.7%) | 3 (12.0%) |
| Before/during/after sex | 0 | 0 | 0 | 0 | 1 (7.7%) | 1 (4.0%) |
| Toileting event[2] | 0 | 2 (16.7%) | 2 (8.7%) | 3 (25.0%) | 5 (38.5%) | 8 (32.0%) |
| Exercising/other activity | 0 | 1 (8.3%) | 1 (4.3%) | 4 (33.3%) | 1 (7.7%) | 5 (20.0%) |
| Other[3] | 0 | 0 | 0 | 1 (8.3%) | 2 (15.4%) | 3 (12.0%) |
| Number of bleeding days, median (range) | 0 (0, 6) | 0 (0, 9) | 0 (0, 9) | 23 (8, 75) | 32 (0, 78) | 30 (0, 78) |
| Bleeding incidence rate (bleeding days per person-month), 95% CI | 1.3 | 3.7 | 2.6 | 12.9 | 12.8 | 12.8 |
| | (0.6, 2.5) | (2.4, 5.5) | (1.8, 3.6) | (11.7, 14.2) | (11.7, 14.0) | (12.0, 13.7) |

1. Participants without at least one full outage (fully fell out or removed) are considered adherent. MTN-044 participants reported partial outage(s) but only participants with no full outages are considered adherent. All outages reported on MTN-030 Ring Adherence CRF were full outages.

2. Bowel movement/urination/ring falling in toilet

3. Reasons for ring outages collected by SMS and "Other" reasons not specified further.

continuous arm. Vaginal bleeding incidence (bleeding days per person-month) was comparable between the continuous (12.9; 95% CI: 11.7, 14.2) and cyclic (12.8; 95% CI: 11.7, 14.0) use arms (Table 3).

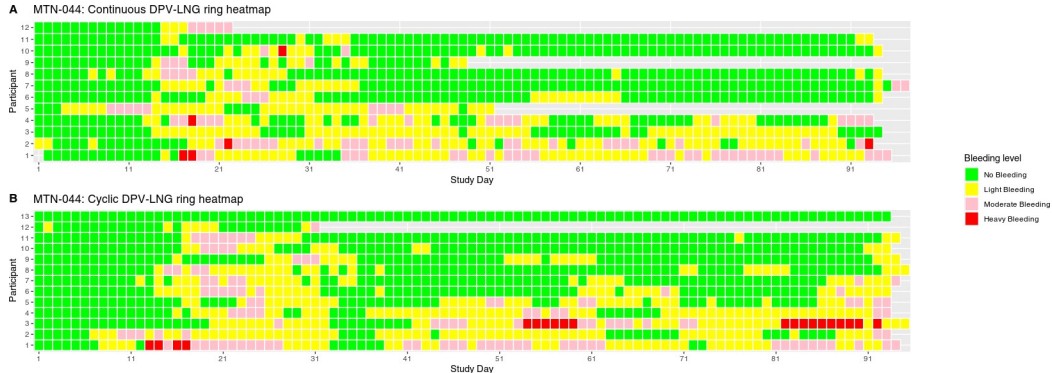

**Fig 5. Heatmap of bleeding patterns, MTN-044/IPM 053/CCN019 (n = 25).** green = no bleeding, yellow = light bleeding, pink = moderate bleeding, red = heavy bleeding.

## Adherence

In MTN-030, 20/23 women were adherent (ring in place for the entire 14 days). Three women had a ring outage, all lasting 15 minutes or less. All three outages were involuntary expulsions in the DPV-LNG ring arm, one of which overlapped with a day of light, non-menstrual bleeding. In MTN-044, a total of 20 women experienced any outages, including partial or complete expulsions or unscheduled removals (Table 3). The 13/25 women who experienced no full outages were considered adherent, with no differences by arm. Among women with full outages (due to expulsions or removals), the most common reasons cited were the ring being out of place or moving (75%); bowel movement, urination or ring falling in the toilet (67%); and exercise or other activities (42%), with no significant differences by arm.

## Factors associated with acceptability

In MTN-030, we found no significant associations between product characteristics, adherence, or vaginal changes (such as increased wetness or dryness), and overall acceptability (Table 4). In MTN-044, we found no significant association between adherence and acceptability (Fisher's Exact Test, p = 0.43). Although not statistically significant (Fisher's Exact Test, p = 0.43), more women who liked the ring were adherent (8/13) than those who disliked the ring (5/12). In addition, women who disliked the ring did not have significantly more outages than those who liked the ring (median 1.5, range 0–39 versus median 1.0, range 0–5; logistic regression p = 0.21). However, more women who disliked (7/12) versus liked (1/13) the ring attributed outages to involuntary expulsions during toileting—bowel movement, urination or ring falling in toilet (Fisher's Exact Test, p = 0.01). Women who disliked the ring (in both arms) had a significantly higher incidence of vaginal bleeding than those who liked the ring (17.8 bleeding days per person-months, 95% CI 16.2, 19.3 versus 8.8 bleeding days per person-months, 95% CI: 7.9, 9.8). Furthermore, significantly more women who disliked the ring reported that vaginal bleeding changes were unacceptable than those who liked it (7/12 versus 0/13; Fisher's Exact Test: unadjusted two-sided p-value = 0.002). One woman who said the bleeding changes were unacceptable reported approximately 67 days of bleeding and was found to have expelled the ring at an unknown date (between Days 74 and 90) prior to her Day 90/end of product use visit. In univariate analyses, we found that women who had used a condom at last sex at baseline (versus did not use a condom) had significantly lower odds of liking the ring after completing product use (OR 0.1, 95% CI: 0.1, 0.6; p = 0.01), as did women who were not worried about wearing the ring versus worried about wearing the ring (OR 0.1, 95% CI: 0.1, 0.6; p = 0.01). By contrast, we found that high school graduates had significantly higher odds of liking the ring compared to college graduates (OR 12.8, 95% CI: 1.3, 130.5; p = 0.03). Women with no prior IUD use (versus prior IUD use) or who were single (versus married/in a relationship/divorced) had higher odds of liking the ring (OR 5.5, 95% CI: 0.8, 36.2; p = 0.08 and OR 4.8, 95% CI: 0.9, 26.8; p = 0.07, respectively), but these associations were not significant, and no associations were significant in multivariable analysis.

## Discussion

In these two early Phase 1 trials of the DPV-LNG ring, most women (all MTN-030 participants, 84% of MTN-044 participants) said the ring was easy to use and comfortable to use for the duration of their study participation. However, more participants reported thinking about the ring being in their bodies or being aware of it during daily activities when using it for 90 days in MTN-044 than those using it for only 14 days in MTN-030. Furthermore, most women in MTN-030 liked their assigned ring after using it continuously for 14 days whereas

**Table 4. Impact of product characteristics, adherence, and bleeding patterns on overall acceptability, by study (MTN-030/IPM 041 [n = 23] and MTN-044/IPM 053/CCN019 [n = 25]).**

| Variable | MTN-030/IPM041 | | | MTN-044/IPM 053/CCN019 | | |
|---|---|---|---|---|---|---|
| Overall Acceptability | Like/Like very much n = 21 | Dislike/ Dislike very much n = 2 | Total (n = 23) | Like/Like very much n = 13 | Dislike/ Dislike very much n = 12 | Total (n = 25) |
| **How worried are you about having the ring inside you? (baseline)** | | | | | | |
| Somewhat worried | 1 (5%) | 1 (50%) | 2 (9%) | 5 (38%) | 0 (0%) | 5 (20%) |
| A little worried | 6 (29%) | 0 (0%) | 6 (26%) | 5 (38%) | 3 (25%) | 8 (32%) |
| Not at all worried | 14 (67%) | 1 (50%) | 15 (65%) | 3 (23%) | 9 (75%) | 12 (48%) |
| **Overall, how easy or difficult was it to use the ring?** | | | | | | |
| Very easy | 16 (76%) | 1 (50%) | 17 (74%) | 7 (54%) | 6 (50%) | 13 (52%) |
| Easy | 5 (24%) | 1 (50%) | 6 (26%) | 5 (38%) | 3 (25%) | 8 (32%) |
| Difficult | 0 (0%) | 0 (0%) | 0 (0%) | 1 (8%) | 3 (25%) | 4 (16%) |
| **The first time you inserted the ring in your vagina, was it difficult or easy to insert?** | | | | | | |
| Very easy | 4 (19%) | 1 (50%) | 5 (22%) | 3 (23%) | 5 (42%) | 8 (42%) |
| Easy | 9 (43%) | 1 (50%) | 10 (43%) | 6 (46%) | 5 (42%) | 11 (42%) |
| Difficult | 5 (24%) | 0 (0%) | 5 (22%) | 4 (31%) | 2 (17%) | 6 (17%) |
| I did not insert ring | 3 (14%) | 0 (0%) | 3 (13%) | 0 (0%) | 0 (0%) | 0 (0%) |
| **Overall, how often did you think about the ring being inside your body?** | | | | | | |
| Never | 8 (38%) | 0 (0%) | 8 (35%) | 1 (8%) | 2 (17%) | 3 (12%) |
| Ever | 13 (62%) | 2 (100%) | 15 (65%) | 12 (92%) | 10 (83%) | 22 (88%) |
| **Overall, were you aware of the ring during your normal daily activities?** | | | | | | |
| Never | 16 (76%) | 1 (50%) | 17 (74%) | 3 (23%) | 3 (25%) | 6 (24%) |
| Ever | 5 (24%) | 1 (50%) | 6 (26%) | 10 (77%) | 9 (75%) | 19 (76%) |
| **Overall, how did it feel to have the ring inside you every day?** | | | | | | |
| Comfortable | 21 (100%) | 2 (100%) | 23 (100%) | 11 (85%) | 10 (83%) | 21 (84%) |
| Uncomfortable | 0 (0%) | 0 (0%) | 0 (0%) | 2 (15%) | 2 (17%) | 4 (16%) |
| Ever checked to see if the ring was still inside | 5 (24%) | 1 (50%) | 6 (26%) | 7 (54%) | 6 (50%) | 13 (52%) |
| Overall, ever noticed that vagina was wetter | 8 (38%) | 1 (50%) | 9 (39%) | 7 (54%) | 6 (50%) | 13 (52%) |
| **How much has your vagina being wetter bothered you?** | | | | | | |
| Not at all | 2 (25%) | 0 (0%) | 2 (22%) | 4 (57%) | 2 (33%) | 6 (46%) |
| A little | 5 (63%) | 1 (100%) | 6 (67%) | 3 (43%) | 3 (43%) | 6 (46%) |
| Somewhat | 1 (13%) | 0 (0%) | 1 (11%) | 0 (0%) | 0 (0%) | 0 (0%) |
| Very much | 0 (0%) | 0 (0%) | 0 (0%) | 0 (0%) | 1 (17%) | 1 (8%) |
| Overall, ever noticed that vagina was drier | 3 (23%) | 1 (100%) | 4 (29%) | 1 (8%) | 0 (0%) | 1 (4%) |
| **How much has your vagina being drier bothered you?** | | | | | | |
| Not at all | 1 (33%) | 1 (100%) | 2 (50%) | 0 (0%) | 0 (0%) | 0 (0%) |
| A little | 1 (33%) | 0 (0%) | 1 (25%) | 0 (0%) | 0 (0%) | 0 (0%) |
| Somewhat | 1 (33%) | 0 (0%) | 1 (25%) | 1 (100%) | 0 (0%) | 1 (100%) |
| Number of ring outages, median (range) | 0 (0, 1) | 0 (0, 1) | 0 (0, 1) | 0 (0, 5) | 1.5 (0, 39) | 0 (0, 39) |
| Adherent[1] | 19 (90%) | 1 (50%) | 20 (87%) | 8 (62%) | 5 (42%) | 13 (52%) |
| **Study:** | **MTN-030/IPM 041** | | | **MTN-044/IPM 053/CCN019** | | |
| **Reasons for outages** | | | | | | |
| Ring out of place/moved | 1 (5%) | 1 (50%) | 2 (9%) | 5 (38%) | 4 (33%) | 9 (36%) |
| Physical discomfort | 0 (0%) | 0 (0%) | 0 (0%) | 3 (23%) | 1 (8%) | 4 (16%) |
| Menses | 0 (0%) | 0 (0%) | 0 (0%) | 1 (8%) | 2 (17%) | 3 (12%) |
| Before/during/after sex | 0 (0%) | 0 (0%) | 0 (0%) | 1 (8%) | 0 (0%) | 1 (4%) |
| Toileting event[2] | 0 (0%) | 1 (50%) | 1 (4%) | 1 (8%) | 7 (58%) | 8 (32%)* |
| Exercising/other activity | 1 (5%) | 0 (0%) | 0 (0%) | 2 (15%) | 3 (25%) | 5 (20%) |
| Other[3] | 0 (0%) | 0 (0%) | 0 (0%) | 1 (8%) | 2 (17%) | 3 (12%) |

*(Continued)*

**Table 4.** (Continued)

| Variable | MTN-030/IPM041 | | | MTN-044/IPM 053/CCN019 | | |
|---|---|---|---|---|---|---|
| **Overall Acceptability** | **Like/Like very much n = 21** | **Dislike/ Dislike very much n = 2** | **Total (n = 23)** | **Like/Like very much n = 13** | **Dislike/ Dislike very much n = 12** | **Total (n = 25)** |
| Overall Acceptability: How much do you like the ring overall? | Like/Like very much | Dislike/ Dislike very much | Total (n = 23) | Like/Like very much | Dislike/ Dislike very much | Total (n = 25) |
| | n = 21 (91%) | n = 2 (9%) | | n = 13 (52%) | n = 12 (48%) | |
| Overall, how acceptable were any changes in your bleeding pattern while using the ring?[4] | | | | | | |
| Acceptable[5] | N/A | N/A | NA | 13 (100%) | 5 (42%) | 18 (72%)* |
| Unacceptable | | | | 0 (0%) | 7 (58%) | 7 (28%) |
| Number of Bleeding Days, median (range) | 0 (0, 9) | 0 (0, 0) | 0 (0, 9) | 21 (0, 72) | 53 (8, 78) | 30 (0, 78) ** |
| Bleeding Incidence rate (bleeding days per person-month), 95% CI | 2.8 | 0.0 | 2.6 | 8.8 | 17.8 | 12.8 |
| | (1.9, 3.9) | (0.0, 3.4) | (1.8, 3.6) | (7.9, 9.8) | (16.2, 19.3) | (12.0, 13.7) |

1. Participants without at least one full outage (fully fell out or removed) are considered adherent. MTN-044 participants reported partial outage(s) but only participants with no full outages are considered adherent. All outages reported on MTN-030 Ring Adherence CRF were full outages.

2. Bowel movement/urination/ring falling in toilet

3. Reasons for ring outages collected by SMS and "Other" reasons not specified further.

4. Question was not included in the questionnaire for MTN-030.

5. "Acceptable" includes those who reported no change in bleeding pattern.

* Significant difference between like/like very much and dislike/dislike very much. Fisher's Exact Test, p<0.05

** Significant difference between like/like very much and dislike/dislike very much. Logistic regression p<0.05

only about half of those in MTN-044 liked the DPV-LNG ring after using it continuously or cyclically for up to 90 days.

The different outcomes are not unexpected given the different durations of ring use and comparison groups, study designs and objectives, procedures, and data collection methods. In MTN-030, half of the participants used the DPV-LNG ring while the other half used the DVR-200mg for 14 days continuously. In MTN-044, all participants used the DPV-LNG ring for 90 days, either continuously or cyclically. By design, study visits and ring use were scheduled when menstrual bleeding was not anticipated in MTN-030, whereas in MTN-044, women were instructed to use the ring continuously or cyclically per protocol, even during bleeding episodes.

We found no associations between any variables and overall acceptability in MTN-030, which is not surprising since only two women reported not liking the ring. Although the incidence of vaginal bleeding was not significantly associated with acceptability in MTN-030, the two women who disliked the ring, both in the DPV-LNG ring arm, experienced unanticipated bleeding. It is also notable that the three women who experienced an expulsion were all in the DPV-LNG ring arm. Finally, only 50% of women (6/12) assigned to use the DPV-LNG ring arm said they would be willing to use the ring in the future compared to 64% in the DVR-200mg arm. Although we did not ask participants why they would not want to use the ring, we can hypothesize from their comments that they did not consider themselves to be at risk of HIV (or pregnancy), or that specific product attributes–such as size, compression strength, or durometer (hardness)–or vaginal changes that were not bothersome for 14 days might limit enthusiasm for longer use.

The trends seen in MTN-030 were more pronounced in MTN-044, where higher incidence of vaginal bleeding and unacceptable bleeding changes were the strongest predictors of low acceptability. Although there was no statistically significant difference in frequency of outages

overall between those who liked or disliked the ring, most participants (19/25) experienced an expulsion, and women who disliked the ring experienced more expulsions than those who liked the ring. Furthermore, significantly more women who disliked the ring experienced expulsions during toileting compared to women who liked the ring. As noted above, in MTN-044, willingness to use the ring in the future was discussed in in-depth interviews and is described in detail elsewhere [38]. In brief, of the 17 women in MTN-044 who said they would be willing to use the ring in the future, only three said they would use it as is, with no changes. The other 14 women, many of whom had experienced unexpected vaginal bleeding, feeling like the ring was slipping, or complete expulsions, said that if they were at higher risk of HIV, the benefits would outweigh the interference in their daily activities.

The DPV-LNG ring was well-tolerated and achieved target drug levels needed for efficacy. However, the results of this study suggest that ring expulsions and bleeding irregularities decreased the overall acceptability of this product. Our findings have been critical for redesigning the DPV-LNG ring when it is still feasible to make significant changes. To that end, the product developers evaluated rings with different polymers, mechanical properties, such as durometer (hardness), and designs for optimal drug release to limit episodes of non-menstrual bleeding and reduce the likelihood of expulsions. The reformulated DPV-LNG ring is currently being tested in a Phase 1 trial [45].

Early end-user feedback is critical to ensure that MPTs are not only safe and efficacious but have characteristics that facilitate uptake and sustained use [32, 46–52]. Several examples from contraceptive ring development highlight the importance of incorporating end user data early as once products are in efficacy trials, it is too late to change the design if issues related to vaginal bleeding, ease of use, comfort and expulsions indicate acceptability concerns [53–58]. In a study of a levonorgestrel ring designed for 90 days of continuous use, about 17% of women discontinued the trial early due to menstrual disturbances [58]. In clinical trials of the one-month contraceptive NuvaRing®, 30% of participants discontinued use due to discomfort during sex, feeling a "foreign body," or spontaneous (involuntary) expulsions [59]. And in the Phase 3 trial of the one-year contraceptive vaginal system Annovera®, women who removed the ring for longer than two hours were nearly twice as likely to discontinue using it [60]. Several recent reviews of ring studies have highlighted the associations between vaginal bleeding changes, ease of use, expulsions, and impact on sex with acceptability or discontinuation [61–64], yet little empirical data exist on how specific physical ring properties influence the incidence of expulsions and removals and ultimately satisfaction [65].

Research is needed to explore the intersection of ring characteristics and individual women's physical characteristics (such as body mass index, vaginal length and genital hiatus) to improve ring design. Age, partnership status, risk perception, and previous contraceptive experience are likely to have an impact on acceptability, intention to use, uptake, and effective use of the DPV-LNG ring [66]. Although we found no significant associations between women's background characteristics and acceptability of the ring, we did see trends in MTN-044 that will need to be further explored in larger, longer studies. For example, women who had used a condom at last sex or who had previously used an IUD were less likely to find the ring acceptable, perhaps because they were satisfied with their current contraceptive methods. On the other hand, women who had been worried about wearing the ring at baseline were more likely to find the ring acceptable than those who had not been worried about using it. This interesting finding could be about the ring itself, which exceeded expectations for some women and did not meet expectations for others. Or it is possible that some women are more likely to worry about trying a new product than others. Our data reinforce the importance of developing an array of MPTs, as there is no single product that will be right for all individuals.

Our analysis had several limitations. First, both studies were Phase I trials with small sample sizes that had several analytic implications: for some relationships of interest sample size (and limited variability in response) precluded statistical testing whereas for others the sample size limited the ability to estimate associations with precision. In addition, because of the different study designs, rings used (DVR-200mg and DPV-LNG in MTN-030 vs DPV-LNG only in MTN-044), durations (14 versus 90 days) and conditions (with or without sex and menses), and the way data were collected (questions asked in one trial but not in the other), we were unable to combine data from the two trials to conduct more robust analyses. Despite these design constraints, we were able to explore associations that informed design modifications. Second, data were self-reported. If women underreported instances of vaginal bleeding or ring outages, it may mean the ring was less acceptable than our data show. However, because participants had to be at low risk of HIV and using an effective, non-hormonal contraceptive to be in the trials, it is conceivable that the ring would be more acceptable to women at higher risk of HIV or not already using contraception. A third limitation was inclusion of only US women. Additional studies will be needed to assess the acceptability of a multipurpose ring among likely intended users.

In conclusion, the DPV-LNG ring has the potential to increase women's options for simultaneous prevention from HIV and unintended pregnancy. Particularly for prevention products, in which use is a choice (versus products that are necessary and desired to treat or cure an existing medical condition), products must be highly acceptable to users. Gathering acceptability data early in product development is critical for identifying factors that could undermine product uptake and satisfaction. Reformulating the DPV-LNG ring to address vaginal bleeding issues and expulsions is likely to enhance its acceptability for a broader group of users.

## Acknowledgments

We would like to thank the participants and study staff at University of Alabama at Birmingham and University of Pittsburgh for their participation in and implementation of the studies; Irene Bruce for assistance with citations and formatting the manuscript; and Helen Lu for creating the heatmaps.

## Author Contributions

**Conceptualization:** Barbara A. Friedland, Barbara Mensch, Ariane van der Straten, Sharon L. Hiller.

**Data curation:** Holly Gundacker, Clifton W. Kelly.

**Formal analysis:** Holly Gundacker, Clifton W. Kelly.

**Funding acquisition:** Jeanna Piper, Diana Blithe, Sharon L. Hiller.

**Investigation:** Sharon L. Achilles, Beatrice A. Chen, Craig Hoesley, Barbra A. Richardson, John Steytler, Jill Brown.

**Methodology:** Barbara A. Friedland, Sharon L. Achilles, Clifton W. Kelly, Mei Song, Barbara Mensch, Ariane van der Straten, Craig W. Hendrix, Sharon L. Hiller.

**Project administration:** Sharon L. Achilles, Jeanna Piper, Sherri Johnson, Mei Song, Diana Blithe.

**Resources:** Brid Devlin, Kyle Kleinbeck, Bindi Dangi, Chantél Friend, Cindy Jacobson.

**Supervision:** Sharon L. Achilles, Beatrice A. Chen, Craig Hoesley, Barbra A. Richardson, Brid Devlin, Diana Blithe, Sharon L. Hiller.

**Writing – original draft:** Barbara A. Friedland, Holly Gundacker.

**Writing – review & editing:** Barbara A. Friedland, Holly Gundacker, Sharon L. Achilles, Beatrice A. Chen, Craig Hoesley, Barbra A. Richardson, Clifton W. Kelly, Jeanna Piper, Sherri Johnson, Brid Devlin, John Steytler, Kyle Kleinbeck, Bindi Dangi, Chantél Friend, Mei Song, Barbara Mensch, Ariane van der Straten, Cindy Jacobson, Craig W. Hendrix, Jill Brown, Diana Blithe, Sharon L. Hiller.

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
