## [Decision Letter · Decision Letter 0]

1 Apr 2024

PONE-D-23-39000Acceptability of a dapivirine levonorgestrel vaginal ring in two Phase 1 trials (MTN-030/IPM 041 and MTN-044/IPM 053/CCN019): implications for multipurpose prevention technology developmentPLOS ONE

Dear Dr. Friedland,

Thank you for submitting your manuscript to PLOS ONE. After careful consideration, we feel that it has merit but does not fully meet PLOS ONE’s publication criteria as it currently stands. Therefore, we invite you to submit a revised version of the manuscript that addresses the points raised during the review process.

We look forward to receiving your revised manuscript.

Kind regards,

José das Neves

Academic Editor

PLOS ONE

Journal Requirements:

“Brid Devlin, John Steytler, Kyle Kleinbeck, Bindi Dangi, and Chantél Friend are currently employed by or previously employed by the International Partnership for Microbicides (IPM) that developed the dapivirine levonorgestrel ring. Barbara A. Friedland is employed by Population Council, which acquired IPM's assets in 2022.”

We note that one or more of the authors are employed by a commercial company: International Partnership for Microbicides

4. In the online submission form, you indicated that [Data can be shared upon request.].

5. PLOS requires an ORCID iD for the corresponding author in Editorial Manager on papers submitted after December 6th, 2016. Please ensure that you have an ORCID iD and that it is validated in Editorial Manager. To do this, go to ‘Update my Information’ (in the upper left-hand corner of the main menu), and click on the Fetch/Validate link next to the ORCID field. This will take you to the ORCID site and allow you to create a new iD or authenticate a pre-existing iD in Editorial Manager. Please see the following video for instructions on linking an ORCID iD to your Editorial Manager account: https://www.youtube.com/watch?v=_xcclfuvtxQ.

6. Please be informed that funding information should not appear in the Acknowledgments section or other areas of your manuscript. We will only publish funding information present in the Funding Statement section of the online submission form. Please remove any funding-related text from the manuscript.

7. We note that you have included the phrase “data not shown” in your manuscript. Unfortunately, this does not meet our data sharing requirements. PLOS does not permit references to inaccessible data. We require that authors provide all relevant data within the paper, Supporting Information files, or in an acceptable, public repository. Please add a citation to support this phrase or upload the data that corresponds with these findings to a stable repository (such as Figshare or Dryad) and provide and URLs, DOIs, or accession numbers that may be used to access these data. Or, if the data are not a core part of the research being presented in your study, we ask that you remove the phrase that refers to these data.

Reviewers' comments:

Reviewer's Responses to Questions

**Comments to the Author**

1. Is the manuscript technically sound, and do the data support the conclusions?

Reviewer #1: Partly

Reviewer #2: Yes

2. Has the statistical analysis been performed appropriately and rigorously? 

Reviewer #1: Yes

Reviewer #2: Yes

3. Have the authors made all data underlying the findings in their manuscript fully available?

Reviewer #1: Yes

Reviewer #2: Yes

4. Is the manuscript presented in an intelligible fashion and written in standard English?

Reviewer #1: Yes

Reviewer #2: Yes

5. Review Comments to the Author

Reviewer #1: This manuscript presents acceptability data on the dapivirine levonorgestrel (DPV-LNG) ring from two randomized trials of non-pregnant cisgender women without HIV at the University of Pittsburgh and University of Alabama. Overall, most participants (n=21/23) in a randomized study offering DPV-LNG versus solely DVR for 14 days liked the ring modality they were offered, with no significant difference in acceptability between arm offered DPV-LNG versus DVR-only ring. About half (n=13/25) in the study offering DPV-LNG for 90 days liked the DPV-LNG ring. Those who disliked the ring in MTN-044 reported significantly higher incidence of unanticipated vaginal bleeding, and ring expulsion was reported more frequently by women who disliked the DPV-LNG ring.

The overall study is well-conducted, and the authors have identified an important gap in understanding factors impacting combination ring acceptability, adherence, and persistence. The methodology is straightforward, and the results are clearly described. My feedback is described below in detail.

Major revision:

1. The described objective of the paper is to explore acceptability of the DPV-LNG ring. However, the acceptability data from MTN-030 presented in Table 3 combines data from both the DVR and DPV-LNG arms, despite much of the data interpretation related to the attributes discussed in Table 3 being separated by arm in the text (lines 309-341). Further, the abstract of the paper asserts that 21/23 women liked the DPV-LNG ring in lines 62-63, when this from my understanding is referencing combined acceptability of both ring methods offered in MTN-030. In addition to modifying this data interpretation in the abstract, would it be possible to make this clearer in Table 3, that MTN-030 is presenting acceptability of both DVR/DPV-LNG, whereas MTN-044 is presenting acceptability of only DPV-LNG options? Alternatively, would the author consider presenting solely the DPV-LNG data from MTN-030 in Table 3?

Minor revision:

1. Could you please define adherence and persistence as initially presented in line 79?

2. Lines 112-113: Was reporting effective use of condoms considered use of an effective, non-hormonal contraceptive?

3. Could you specify which of the measures described in lines 185 through 197 would measure persistence on the DPV-LNG ring, as opposed to adherence, per the above requested definitions of those terms?

Reviewer #2: Manuscript is well written and commend the researchers on this great work that is important to inform future development of MPTs.

Just minor revision on line 240, there is a punctuation sign missing after the words "due to" as it makes reading of reasons for 4 expulsions confusing when missing.

I was struck by the expulsions in MTN044 among women who experienced vaginal bleeding however good enough the authors mention data on expulsions was based off participant report as a limitation. Women experiencing these bleedings could intentionally remove the rings and report this as unintentional expulsion.

Did the participants have ring placement confirmed by attending study staff after first insertion to ensure it is in right place?

6. PLOS authors have the option to publish the peer review history of their article (what does this mean?). If published, this will include your full peer review and any attached files.

Reviewer #1: No

Reviewer #2: No

---

## [Author Response · Author response to Decision Letter 0]

25 Sep 2024

1. Please ensure that your manuscript meets PLOS ONE's style requirements. DONE

2. Thank you for stating the following in the Competing Interests section: “Brid Devlin, John Steytler, Kyle Kleinbeck, Bindi Dangi, and Chantél Friend are currently employed by or previously employed by the International Partnership for Microbicides (IPM) that developed the dapivirine levonorgestrel ring. Barbara A. Friedland is employed by Population Council, which acquired IPM's assets in 2022.” We note that one or more of the authors are employed by a commercial company: International Partnership for Microbicides. THE POPULATION COUNCIL AND IPM ARE NOT COMMERCIAL COMPANIES. WE HAVE EXPLAINED THIS IN THE AMENDED FUNDING STATEMENT IN THE COVER LETTER.

a. Please provide an amended Funding Statement declaring this commercial affiliation, as well as a statement regarding the Role of Funders in your study. If the funding organization did not play a role in the study design, data collection and analysis, decision to publish, or preparation of the manuscript and only provided financial support in the form of authors' salaries and/or research materials, please review your statements relating to the author contributions, and ensure you have specifically and accurately indicated the role(s) that these authors had in your study. You can update author roles in the Author Contributions section of the online submission form. N/A -- NOT COMMERCIAL ENTITY

Please also include the following statement within your amended Funding Statement. WE HAVE AMENDED THE FUNDING STATEMENT TO MATCH EXACTLY WHAT WAS IN THE MAIN PAPER, ALSO PUBLISHED BY PLOS ONE.

“The funder provided support in the form of salaries for authors [insert relevant initials], but did not have any additional role in the study design, data collection and analysis, decision to publish, or preparation of the manuscript. The specific roles of these authors are articulated in the ‘author contributions’ section.” ALREADY INCLUDED.

If your commercial affiliation did play a role in your study, please state and explain this role within your updated Funding Statement. N/A -- NO COMMERCIAL AFFILIATION 

Please include both an updated Funding Statement and Competing Interests Statement in your cover letter. We will change the online submission form on your behalf. DONE

When you resubmit, please ensure that you provide the correct grant numbers for the awards you received for your study in the ‘Funding Information’ section. DONE -- WE HAVE CHECKED AND THE FUNDING INFORMATION MATCHES WHAT WAS IN THE PRIMARY PUBLICIATION. 

4. In the online submission form, you indicated that [Data can be shared upon request.].

THE DATA HAVE NOW BEEN UPLOADED TO A PUBLIC REPOSITORY AND THE DOI WILL BE AVAILABLE BEFORE THE PAPER IS PUBLISHED.

5. PLOS requires an ORCID iD for the corresponding author in Editorial Manager on papers submitted after December 6th, 2016. Please ensure that you have an ORCID iD and that it is validated in Editorial Manager. To do this, go to ‘Update my Information’ (in the upper left-hand corner of the main menu), and click on the Fetch/Validate link next to the ORCID field. This will take you to the ORCID site and allow you to create a new iD or authenticate a pre-existing iD in Editorial Manager. Please see the following video for instructions on linking an ORCID iD to your Editorial Manager account: https://www.youtube.com/watch?v=_xcclfuvtxQ.

I DO HAVE AN ORCID ID BUT IT IS NOT ALLOWING ME TO LINK -- I WILL CONTINUE TO WORK ON THAT. MY ID IS 0000-0002-5041-3634. I DO NOT WANT TO HOLD UP THE SUBMISSION FURTHER.

6. Please be informed that funding information should not appear in the Acknowledgments section or other areas of your manuscript. We will only publish funding information present in the Funding Statement section of the online submission form. Please remove any funding-related text from the manuscript. REMOVED.

7. We note that you have included the phrase “data not shown” in your manuscript. Unfortunately, this does not meet our data sharing requirements. PLOS does not permit references to inaccessible data. We require that authors provide all relevant data within the paper, Supporting Information files, or in an acceptable, public repository. Please add a citation to support this phrase or upload the data that corresponds with these findings to a stable repository (such as Figshare or Dryad) and provide and URLs, DOIs, or accession numbers that may be used to access these data. Or, if the data are not a core part of the research being presented in your study, we ask that you remove the phrase that refers to these data. REMOVED DATA NOT SHOWN AND ADDED DATA ON PAGE 32.

Reviewers' comments:

Reviewer's Responses to Questions

Comments to the Author

1. Is the manuscript technically sound, and do the data support the conclusions?

Reviewer #1: Partly

Reviewer #2: Yes

2. Has the statistical analysis been performed appropriately and rigorously?

Reviewer #1: Yes

Reviewer #2: Yes

3. Have the authors made all data underlying the findings in their manuscript fully available?

Reviewer #1: Yes

Reviewer #2: Yes

4. Is the manuscript presented in an intelligible fashion and written in standard English?

Reviewer #1: Yes

Reviewer #2: Yes

5. Review Comments to the Author

Reviewer #1: This manuscript presents acceptability data on the dapivirine levonorgestrel (DPV-LNG) ring from two randomized trials of non-pregnant cisgender women without HIV at the University of Pittsburgh and University of Alabama. Overall, most participants (n=21/23) in a randomized study offering DPV-LNG versus solely DVR for 14 days liked the ring modality they were offered, with no significant difference in acceptability between arm offered DPV-LNG versus DVR-only ring. About half (n=13/25) in the study offering DPV-LNG for 90 days liked the DPV-LNG ring. Those who disliked the ring in MTN-044 reported significantly higher incidence of unanticipated vaginal bleeding, and ring expulsion was reported more frequently by women who disliked the DPV-LNG ring.

The overall study is well-conducted, and the authors have identified an important gap in understanding factors impacting combination ring acceptability, adherence, and persistence. The methodology is straightforward, and the results are clearly described. My feedback is described below in detail.

Major revision:

1. The described objective of the paper is to explore acceptability of the DPV-LNG ring. However, the acceptability data from MTN-030 presented in Table 3 combines data from both the DVR and DPV-LNG arms, despite much of the data interpretation related to the attributes discussed in Table 3 being separated by arm in the text (lines 309-341). Further, the abstract of the paper asserts that 21/23 women liked the DPV-LNG ring in lines 62-63, when this from my understanding is referencing combined acceptability of both ring methods offered in MTN-030. In addition to modifying this data interpretation in the abstract, would it be possible to make this clearer in Table 3, that MTN-030 is presenting acceptability of both DVR/DPV-LNG, whereas MTN-044 is presenting acceptability of only DPV-LNG options? Alternatively, would the author consider presenting solely the DPV-LNG data from MTN-030 in Table 3?

We thank the reviewer for this comment. We discussed the relative merits of including vs excluding the DVR arm from MTN-030, but felt it was important to keep as it was the comparator in the MTN-030 trial. However, we have made several significant revisions to better articulate the differences in opinions about the DVR vs the DVR-LNG rings within both the text and the tables. The specific revisions we made to address these issues are as follows:

• We added a new table (Table 2) displaying acceptability ratings by randomization group. 

• We then reorganized the results to present the acceptability by randomization group per study first (the new Table 2), followed by data on adherence and vaginal bleeding by randomization group. 

• Table 4 is an updated version of the original Table 3, which displays impact of product characteristics, adherence and bleeding on acceptability (like/dislike) of the rings. 

• We also revised the abstract to make it clear when we are talking about either of the assigned rings vs the DPV-LNG ring.

Minor revision:

1. Could you please define adherence and persistence as initially presented in line 79?

Thank you for this comment. We define adherence later in the manuscript in the methods section, however, because we did not measure persistence in this trial, we have removed reference to persistence in the introduction. 

2. Lines 112-113: Was reporting effective use of condoms considered use of an effective, non-hormonal contraceptive?

Thank you for this comment. Condoms were not considered an effective method in the 14-day MN-030/IPM 041 study. We have clarified this in the manuscript on page 5, line 12.

3. Could you specify which of the measures described in lines 185 through 197 would measure persistence on the DPV-LNG ring, as opposed to adherence, per the above requested definitions of those terms?

As noted above, we removed reference to persistence which was not measured in this study.

Reviewer #2: Manuscript is well written and commend the researchers on this great work that is important to inform future development of MPTs.

Just minor revision on line 240, there is a punctuation sign missing after the words "due to" as it makes reading of reasons for 4 expulsions confusing when missing.

We thank the reviewer for these comments. 

We have edited the text for clarity (lines 243-247)

I was struck by the expulsions in MTN044 among women who experienced vaginal bleeding however good enough the authors mention data on expulsions was based off participant report as a limitation. Women experiencing these bleedings could intentionally remove the rings and report this as unintentional expulsion.

We thank the reviewer for this comment and acknowledge that reliance on self-report has the potential to lead to an underreporting of expulsions/non-adherence, whether due to bleeding or other reasons. We mention issues with relying on self-report in the discussion section.

Did the participants have ring placement confirmed by attending study staff after first insertion to ensure it is in right place?

Thank you for this question. We have added in the manuscript that if the participant inserted the ring herself, placement was checked by a clinician (lines 122-123; 143-144).

6. PLOS authors have the option to publish the peer review history of their article (what does this mean?). If published, this will include your full peer review and any attached files.

Do you want your identity to be public for this peer review? For information about this choice, including consent withdrawal, please see our Privacy Policy.

Reviewer #1: No

Reviewer #2: No

While revising your submission, please upload your figure files to the Preflight Analysis and Conversion Engine (PACE) digital diagnostic tool, https://pacev2.apexcovantage.com/. PACE helps ensure that figures meet PLOS requirements. To use PACE, you must first register as a user. Registration is free. Then, login and navigate to the UPLOAD tab, where you will find detailed instructions on how to use the tool. If you encounter any issues or have any questions when using PACE, please email PLOS at <a href="mailto:figures@plos.org">figures@plos.org. Please note that Supporting Information files do not need this step.

---

## [Editor Report · Decision Letter 1]

16 Oct 2024

Acceptability of a dapivirine levonorgestrel vaginal ring in two Phase 1 trials (MTN-030/IPM 041 and MTN-044/IPM 053/CCN019): implications for multipurpose prevention technology development

PONE-D-23-39000R1

Dear Dr. Friedland,

We’re pleased to inform you that your manuscript has been judged scientifically suitable for publication and will be formally accepted for publication once it meets all outstanding technical requirements.

Kind regards,

José das Neves

Academic Editor

PLOS ONE
---

## [Editor Report · Acceptance letter]

20 Nov 2024

PONE-D-23-39000R1 

PLOS ONE

Dear Dr. Friedland, 

I'm pleased to inform you that your manuscript has been deemed suitable for publication in PLOS ONE. Congratulations! Your manuscript is now being handed over to our production team.

Kind regards, 

on behalf of

Dr. José das Neves 

Academic Editor

PLOS ONE